# Hospital unit working conditions and risk for employee injury

Emrah Gecili[1,2]*, Nancy Daraiseh[1,2], Cole Brokamp[1,2], Maurizio Macaluso[1,2]

**1** Division of Biostatistics & Epidemiology, Cincinnati Children's Hospital Medical Center, Cincinnati, Ohio, United States of America, **2** Department of Pediatrics, University of Cincinnati, Cincinnati, Ohio, United States of America

* emrah.gecili@cchmc.org

## Abstract

### Background

Healthcare employees, particularly in pediatric hospitals, are at high risk of occupational injuries. However, few studies have examined hospital unit-level factors that contribute to these injuries. This study aimed to explore and quantify such risk factors in a large pediatric inpatient setting.

### Methods

We conducted a secondary analysis of prospectively collected data for about 7,929 unit-days between 2014 and 2017. Data sources included an institutional injury surveillance system, incident reports collected from a sample of employees through active surveillance (voice recording), and hospital unit-based measures of patient density and employee workload.

Potential determinants of injury among hospital employees included shift length distributions, staffing metrics, patient aggression, near-miss events, and prior injuries. Mixed-effects logistic regression models with unit-level random intercepts were used to evaluate the association between unit-level risk factors and the odds that employee injuries were reported on a unit on a given day.

### Results

Shifts exceeding 13 hours on the previous day were associated with 3–4% higher odds of injury, while same-day shifts shorter than 8.5 hours were associated with a 1% reduction in injury odds. Patient aggression was identified as a significant predictor, greatly increasing the risk of injury, but the association was no longer statistically significant after adjusting for prior injuries. Prior week injuries remained a strong and consistent predictor of future injury occurrences. Near-misses detected in the past

**Data availability statement:** The data that support the findings of this study contain potentially identifying and sensitive employee information and are therefore subject to institutional privacy restrictions. Data access is governed by the Cincinnati Children's Hospital Medical Center (CCHMC) Institutional Review Board (IRB). Qualified researchers may request access to de-identified data through the CCHMC IRB (contact: irb@cchmc.org) pending institutional approval and a data use agreement.

**Funding:** This work was funded by the Centers of Disease Control and Prevention and National Institute for Occupational Safety and Health (ND, 1R21OH010035-01A1). The sponsors or funders did not play any role in the study design, data collection and analysis, decision to publish, or preparation of the manuscript. (https://www.cdc.gov/niosh/).

**Competing interests:** The authors declare no conflict of interest in relation to this manuscript.

week were a significant predictor of injury reporting in unadjusted analyses, but the association was not statistically significant after adjusting for other factors.

## Conclusions

Unit-level risk factors—including work shift duration, patient aggression, and prior injury occurrences—play a significant role in employee injury risk. These findings support the importance of continuous monitoring and targeted interventions, such as shift scheduling limits and systematic near-miss reporting, to enhance occupational safety in pediatric hospital settings.

---

## 1. Introduction

Hospitals are dynamic, demanding, and at times hard-to-predict workplaces. Healthcare workers operate in an environment that can be fraught with risks, both physical and psychological. Their occupation, which centers on providing care to the ill and vulnerable, inherently exposes them to a variety of hazards. Despite the critical importance of employee safety in healthcare settings, the specific risk factors contributing to injuries among hospital staff remain underexplored, especially in pediatric care. A recent nationwide study in Turkey reported that nurses and midwives were among the most affected groups by occupational injuries, emphasizing the need for targeted interventions to improve workplace safety in healthcare settings [1].

Occupational injuries have been well documented as a significant concern in healthcare. The U.S. Bureau of Labor Statistics consistently reports higher injury rates in healthcare than in most other industries [2]. Occupational injuries not only affect the well-being of healthcare employees but also have a direct impact on the quality of patient care. Previous research has shown that injured workers are more likely to experience burnout, reduced job satisfaction, and absenteeism, all of which can lead to decreased patient safety [3,4]. A safe working environment promotes the retention of skilled staff, reduces absenteeism due to work-related injuries or illnesses, and minimizes liability for healthcare institutions [1–4]. When healthcare workers feel safe, they are more likely to be engaged, focused, and efficient.

One of the critical risk factors identified in the literature is extended work hours and overtime. Long working hours have been linked to increased fatigue, decreased vigilance, and a higher likelihood of occupational injuries. For instance, Dembe et al. (2005) found that extended work hours significantly increase the risk of occupational injuries [5]. Similarly, multiple studies have demonstrated that shift lengths beyond 12 hours are associated with a higher incidence of work-related accidents, particularly among nurses. Smith et al. (1998) reported greater injury rates associated with longer shift systems [6], while Imes et al. (2023) identified a strong association between long work hours and needlestick injuries [7]. Caruso (2004) observed that shifts exceeding 12.5 hours elevate the risk of errors [8], and Stimpfel et al. (2012) linked shifts longer than 13 hours to increased burnout and reduced performance [9].

Another emerging risk factor contributing to workplace injuries in healthcare settings is patient aggression. Healthcare workers, particularly those in high-stress environments such as emergency departments and psychiatric units, often encounter aggressive behavior from patients, leading to physical injuries. A 2023 nationwide survey by National Nurses United revealed that over 81% of nurses experienced at least one form of workplace violence in the past year, with nearly half reporting an increase in such incidents [10]. Additionally, 73% of all nonfatal workplace injuries due to violence in the U.S. were sustained by healthcare workers [11]. The combination of long work hours, high workloads, and the potential for patient aggression creates a challenging environment that can exacerbate the risk of injury for healthcare workers.

While the general association between work conditions in healthcare and injury risk has been established, there is a paucity of research that specifically examines the unique working conditions within pediatric hospital units. Existing literature often focuses on general hospital environments, overlooking the specific challenges faced by pediatric providers. Moreover, the impacts of overtime, extended shifts, and patient aggression have not been comprehensively studied in these settings. Most prior studies rely solely on survey responses or institutional incident reporting systems, limiting insight into near-miss events and contextual workplace conditions.

We employed existing data to identify modifiable unit-level risk factors associated with staff injuries in a pediatric hospital. Data sources included reports from the institutional injury reporting hotline, which captures staff-reported incidents, and data from a prior study that examined an active, real-time injury reporting mechanism described in (Macaluso et al 2021). By examining staffing patterns, workload metrics, patient aggression, and prior injury events, we seek to inform preventive strategies that enhance workplace safety.

By identifying modifiable working conditions, this study aims to inform actionable interventions that promote occupational safety in pediatric and similar healthcare environments.

## 2. Materials and methods

### 2.1 Data collection

We conducted a retrospective observational study at a large Midwestern pediatric medical center, analyzing data collected from September 1, 2014, to January 31, 2017. The study focused on employee injuries within this specific institutional context. We obtained employee safety reports for 25 pediatric inpatient hospital units, which were grouped into 15 categories by specialty. These categories fell into three broad groups: Intensive Care Units (ICUs), Medical/Surgical (Med/Surg), and Psychiatry units. Employees included registered nurses (RNs), patient care assistants (PCAs), licensed practical nurses (LPNs), and behavioral health specialists (BHSs). Participant sampling and data collection occurred from February 11, 2013, to July 31, 2015, at the same pediatric center. Adult participants provided written informed consent; minors were not included in the study. Units with incomplete data, reports lacking sufficient detail, and employees who did not consent or provided incomplete recordings were excluded from the respective analyses.

The study integrates two complementary data sources: (1) A passive institutional injury surveillance system (IISS) encompassing 25 inpatient units and approximately 1,214 employees, and (2) an active surveillance study involving 607 employees who recorded injury and near-miss events via handheld voice recorders. The passive surveillance dataset contributed 7,929 unit-days of data covering operational and staffing conditions across ICU, medical–surgical, and psychiatry units, while the active surveillance dataset contributed 1,517 unit-days from units participating in the voice recording study. Thus, two relevant sample sizes are reported: the number of unit-days in the passive surveillance dataset and the number of employees in the active surveillance dataset. The analytic unit for all statistical models is the unit-day, representing all staff scheduled to work on a given unit during a given day.

We considered three metrics related to staffing, operations, and workload that hospital management routinely computes for each unit (unit working conditions) as potentially common risk factors for employee harm:

1) Percentage of Occupancy to Budgeted Average Daily Census (%ADC) is computed daily as 100 x the actual patient census divided by the expected (budgeted) value. When the measure exceeds 100%, the unit is understaffed and may request additional resources (e.g., staff from other units).

2) Nurse orientation hours are computed daily as 100 x (Hours spent in orientation by unit nurses/Productive Hours). This measure may indicate decreased skill level or experience and unfamiliarity with the unit; stress and fatigue from adjusting to new routines and workload [3]. A value exceeding 15% is considered indicative of a lower overall skill level or experience on the unit, potentially impacting patient care and safety.

3) Shifts ≥ 13 hours are computed daily as 100 x (number of shifts lasting ≥13 hours)/ total number of shifts. This measure may be an indicator of staff fatigue and possibly increased risk of error. Higher percentages of such extended shifts are associated with increased fatigue-related errors and injuries [12].

Employees call a central phone number to report all safety events (e.g., slip/trips/falls, aggressive patient; overexertion, caught/struck by object, needlesticks), including Occupational Health & Safety Administration (OSHA) recordables defined as "any illness and injury resulting in death, days away from work, restricted work or transfer to another job, medical treatment beyond first aid, or loss of consciousness." [4]. These reports were transcribed by trained call handlers and coded using standardized criteria. Patient aggression was defined per institution policy as any aggressive/violent behaviors to others and verbal aggression/threats/agitation. A near-miss is defined as "an incident that did not reach a staff member" (e.g., a trip but no fall) or "an incident that reached staff but did not cause harm" (e.g., a bite by an aggressive patient that did not cause harm due to use of Kevlar gloves) [13]. Call handlers ask key questions, with each call lasting 10–45 minutes. These reports comprise the IISS, representing passive surveillance.

To enhance the detection of injury-related events and near-misses, we also incorporated data from a time-limited active surveillance study designed to assess the feasibility and effectiveness of an active surveillance method detailed in Macaluso et al 2018 [14]. Random samples of RNs, PCAs, and BHSs in Med/Surg and psychiatry units were selected by study unit and job type. Participants (N = 607) used handheld digital voice recorders (DVRs) to document any injury or near-miss during their shifts for a two week period.

These active surveillance data did not represent a separate subgroup of staff, but rather provided a complementary dataset that we integrated with the institutional injury surveillance system. One rationale for this integration is that active surveillance captures near misses—events not typically reported in passive surveillance—making it critical for understanding the broader context and risk of injury.

Overall, our dataset included IISS-based injury reports and unit-based risk indicators from 09/01/2014 to 01/31/2017, augmented with additional information from employee voice recordings of select employees and work units from active surveillance conducted between 09/01/2014 and 07/31/2015. The integration of the IISS and active injury surveillance datasets was central to our study's objectives. Passive surveillance, or the IISS, while efficient and broad in scope, often underreports injuries due to its reliance on existing records and reporting mechanisms. In contrast, active surveillance involves proactive data collection and tends to capture a more comprehensive and accurate picture of injury occurrences [14].

The outcome variable in our analysis was a binary indicator of whether an injury occurred on a specific unit on a specific day. We used the time series of unit-based indicators to generate the following derived predictors of injury risk: 1) % shifts >13 h on the same day (represents the proportion of shifts exceeding 13 hours on a given day within a unit), 1 day, 2 days, and 3 days before; 2) the number of days in the past week when the ratio of actual/budgeted patient census decreased from the previous day, 3) the number of days in the past week when the % of nurses in training/orientation decreased from the previous day, 4) % of overtime hours in the past week, 5) % shifts between 8.5 & 12 h on the same day, 6) % shifts <8.5 h on the same day, 7) the number of days in the past week when %shifts >13 h increased over the previous day, in other words the number of days in the preceding week during which the % of shifts longer than 13 h was higher than that of the previous day. This variable captures the frequency of days with an increasing trend in extended

shifts, reflecting potential periods of escalating workload or staffing shortages. We used the time series of IISS injury reports to generate the following derived predictors: 8) the number of injuries reported in the past week, 9) any exposure to bodily fluids in the past week (yes/no), and 10) any injury from patient aggression in the past week (yes/no).

The number of near-misses were only included as a predictor in the follow-up analysis to investigate whether active surveillance enhanced the detection of future employee injuries since it was only available for the subset of units that participated in the active surveillance study as aforementioned.

In our regression analyses, we represented shift length variables as percentages to standardize comparisons across units of varying sizes. This approach allowed us to assess the relative prevalence of specific shift durations within each unit, thereby controlling for unit size. Conversely, variables such as "days in the past week," number of reported injuries, and near-misses were treated as integer counts, reflecting their nature as discrete events or time periods. Importantly, because the models were based on unit-level aggregated data rather than individual worker-shift exposures, the shift variables are not directly interpretable as "per additional shift" effects. A full per-shift or per-hour interpretation would require denominators capturing the number of staff at risk per day, which were not available in our dataset. We acknowledge this limitation in the Discussion and note that future work with more granular denominator data could enable more precise interpretation of injury risk at the individual level.

This retrospective cohort study analyzed prospectively collected data from both institutional injury surveillance systems and an active surveillance component. The full dataset included all unit-day records with complete IISS and operational data. The subset with active surveillance predictors included additional variables—such as near-misses—collected via voice recordings [14]. This subset was used to evaluate the added value of enhanced reporting but was not directly compared to the full cohort.

## 2.2  Statistical analysis

We estimated the association between unit-level risk factors and the odds of employee injury using mixed-effects logistic regression models. The outcome variable was a binary indicator of whether an injury occurred on a specific unit on a specific day. All models included a random intercept for each unit to account for clustering of observations within units over time.

Single-predictor models were used to assess the individual association of each predictor with the outcome. Multivariable models simultaneously included all predictors to adjust for potential confounding and to evaluate the independent effect of each factor. The results are presented as odds ratios (ORs) with corresponding 95% confidence intervals (CIs) and a p-value less than 0.05 was considered statistically significant.

Missing data for key predictors were handled using listwise deletion at the unit-day level. Only unit-day records with complete data were included in the modeling datasets to ensure consistency and validity across analyses.

To address potential bias due to the absence of unit- and day-specific denominators necessary to accurately estimate unit- and day-specific risks or odds of injury, we conducted sensitivity analyses by including the number of nurses working on each unit per day as a covariate. This adjustment allowed us to assess whether daily staffing levels influenced the estimated associations.

All statistical analyses were conducted using R version 4.4.0 [15]. Mixed-effects logistic regression models were fit using the *lme4* package. Code is available in the online supplemental file to support reproducibility.

## 2.3  Ethical considerations

This study was approved by the Cincinnati Children's Hospital Medical Center Institutional Review Board (Study IDs: 2018–5277 and 2013–4482). All adult participants provided informed consent. The study adheres to the STROBE (Strengthening the Reporting of Observational Studies in Epidemiology) guidelines [16]. A completed STROBE checklist is included in the supplementary material.

## 3. Results

We excluded 221 observations with missing data from a pool of 8,150 unit-days, leaving an analytic dataset of 7,929 unit-days without active surveillance predictors. The active surveillance dataset contributed 1,517 unit-days derived from 607 employees participating in the voice recording study. The analyses presented in this work employ one of these two datasets. Descriptive statistics for the outcome variable and potential predictors for both study datasets (with and without active surveillance predictors) are provided in Table 1. Categorical variables were calculated as frequencies and percentages, with percentages representing the proportion of each category relative to the total number of observations in the respective dataset. Continuous variables were summarized using means and standard deviations, calculated over the same totals.

The descriptive characteristics of employees and injury events captured through passive surveillance are summarized in Table S1 in S1 File. Most employees were female (84%), with an average age of 33.3 (SD: 9.9) years, and primarily worked as registered nurses (51%) or mental-health specialists (20%). The majority of reported injuries were related to patient aggression (45%) or physical strain (24%). For the active surveillance cohort, demographic and occupational characteristics are summarized in Table S2 in S1 File, adapted from [14]. Most participants were female (86%), White (81%), and under 30 years old (55%), with 72% working as registered nurses.

The injury occurrence rate was slightly higher in the dataset incorporating active surveillance predictors (6.39%) compared to the dataset without these predictors (6.02%). This suggests that the inclusion of active surveillance data, which encompasses near-misses and other detailed incident reports, may capture additional injury occurrences that are otherwise underreported. Furthermore, the proportion of aggressive patient injuries was substantially higher in the active surveillance dataset (29.93%) compared to the dataset without active surveillance predictors (17.99%). This significant increase indicates that active surveillance methods are more effective in detecting incidents related to patient aggression, which are often underreported in passive surveillance systems. Enhanced detection of such events is crucial for developing targeted interventions to improve workplace safety for healthcare workers.

We first examined the unit-based and passive surveillance-based predictors (N = 7,929), which included 13 potential risk factors. In mixed-effects logistic regression analyses evaluating each predictor separately, a 1% increase in the number of shifts lasting less than 8.5 hours on the same day was associated with a 1% decrease in the odds of injury (OR:0.99, 95% CI: 0.980–0.997; p = 0.0056); a 1% point increase in the number of shifts lasting 8.5–12 hours was associated with a 1% increase (OR:1.01, 95% CI:1.002–1.020; p = 0.0148); and 1% increase in shifts >13 hours on the previous day was associated with a 3% increase the odds of injury (OR:1.03, 95% CI: 1.01, 1.06; p = 0.0096). The number of injuries reported in the past week was also associated with increased odds of injury (OR:1.19, 95%CI:1.12–1.27; p < 0.0001). Any injury from patient aggression in the past week (OR:1.57, 95%CI:1.24, 1.98; p = 0.0002) was associated with increased odds of employee injury (Fig 1 and Table S3 in S1 File in the online supplement).

When evaluating all predictors in the same model, only the associations with percentage of shifts >13 hours on the previous day and the number of injuries reported in the past week remained statistically significant after adjusting for all other risk factors (Fig 1 and Table S1 in S1 File online supplement). Based on these results, the number of injuries reported on a unit during the past week (aOR:1.15, 95%CI:1.07–1.24; p = 0.0003) is predictive for injury during a particular day of operation even after adjusting for other risk factors (when other risk factors are fixed). A 1% increase in shifts >13 hours on the previous day (aOR:1.04, 95%CI:1.01–1.07; p = 0.0118) is associated with increased odds of employee injury, when holding other factors fixed.

In the fully adjusted model including a ll predictors, injury from patient aggression in the past week was not statistically significant (OR = 1.20, 95% CI: 0.91–1.58). After excluding the number of injuries reported in the past week from the model (see Table S4 in S1 File, online supplement), the OR for aggressive patient injury increased to 1.53 (95% CI: 1.21–1.95), becoming statistically significant. Conversely, in a model including only the number of injuries in the past week, this variable remained a robust predictor of injury (OR = 1.18, 95% CI: 1.10–1.26). These patterns, together with the strong

**Table 1. Descriptive statistics for datasets with and without active surveillance predictors.** Categorical variables are presented as counts and percentages (%), calculated over the total number of observations in each dataset (n = 7,929 for the dataset without active surveillance; n = 1,517 for the dataset with active surveillance). Continuous variables are presented as mean values with standard deviations (SD), calculated over the same respective totals.

| Variables | Analysis data without active surveillance predictors (n = 7929) | Analysis data with active surveillance predictors (n = 1517) |
|---|---|---|
| Injury occurrence (yes/no) | 477 (6.02%) | 97 (6.39%) |
| % shifts > 13 h (for one additional % point): | | |
| on the same day | 1.56 (19.69) | 1.43 (3.18) |
| 1 day before | 1.56 (3.254) | 1.40 (3.13) |
| 2 days before | 1.57 (3.263) | 1.41 (3.07) |
| 3 days before | 1.57 (3.255) | 1.43 (3.06) |
| N days in the past week when %shifts > 13 h increased over the previous day (per additional day) | 1.609 (1.26) | 1.59 (1.22) |
| % shifts < 8.5 h on the same day (for one additional % point) | 51.86 (24.59) | 49.28 (21.10) |
| % shifts between 8.5 & 12 h in a unit on the same day (for one additional % point) | 42.81 (22.72) | 45.41 (19.89) |
| % of overtime h in the past week (for one additional % point) | 1.76 (1.81) | 1.67 (1.73) |
| N days in the past week when the %ADC decreased from the previous day (per additional day) | 3.15 (1.17) | 3.29 (1.09) |
| N days in the past week when the % of nurse orientation hours decreased from the previous day | 1.65 (1.64) | 1.70 (1.63) |
| N injuries reported in the past week (per additional injury reported) | 0.501 (1.15) | 0.487 (0.943) |
| Any exposure to bodily fluids (no BBP exposure) in the past week (yes/no) | 126 (1.59%) | 42 (2.76%) |
| Aggressive patient Injury (yes/no) | 1427 (17.99%) | 454 (29.93%) |
| N near-misses reported in the past week (per additional injury reported) | – | 0.654 (1.53) |

BBP: Blood-borne pathogen; %ADC: Percentage of occupancy to budgeted average daily census; N: number; h: hour.

correlation between the two variables (r = 0.66), suggest that joint inclusion may lead to over-adjustment and attenuation of the aggression effect. In addition, it is plausible that recent injuries increase the likelihood of aggressive patient behavior, placing aggression on the causal pathway between a high-risk environment and subsequent employee injury. Under this framework, the two predictors cannot be considered fully independent, and mediation analysis may be a more appropriate approach to disentangle direct and indirect effects. Other predictors, such as shifts exceeding 13 hours, showed relatively stable ORs across all models.

To study risk factors identified through active surveillance, we restricted the analysis to the subset of observations where the sample-based voice recordings were available (N = 1,517). In this case, the list of potential predictors includes the number of near-misses reported in the past week and another binary predictor for aggressive patient injury in the past week, obtained by combining passive and active surveillance data. The average number of near-misses reported in the past week was 0.65 (SD 1.53) per unit in the active surveillance dataset. In mixed effects logistic regression analyses evaluating each predictor separately, the number of days in the past week when the % nurse orientation hours decreased from the previous day (OR:0.81, 95%CI:0.69–0.94; p = 0.0069) was associated with decreased odds of injury while an increased number of injuries reported in the past week (OR:1.37, 95%CI:1.17–1.60; p < 0.0001) was associated with increased odds of injury. Finally, the number of near-misses detected by active surveillance on a unit during the past week (OR:1.15, 95%CI:1.04, 1.27); p = 0.0049) were predictive for injury during a particular work day (Fig 2 and Table S5 in S1 File in the online supplement).

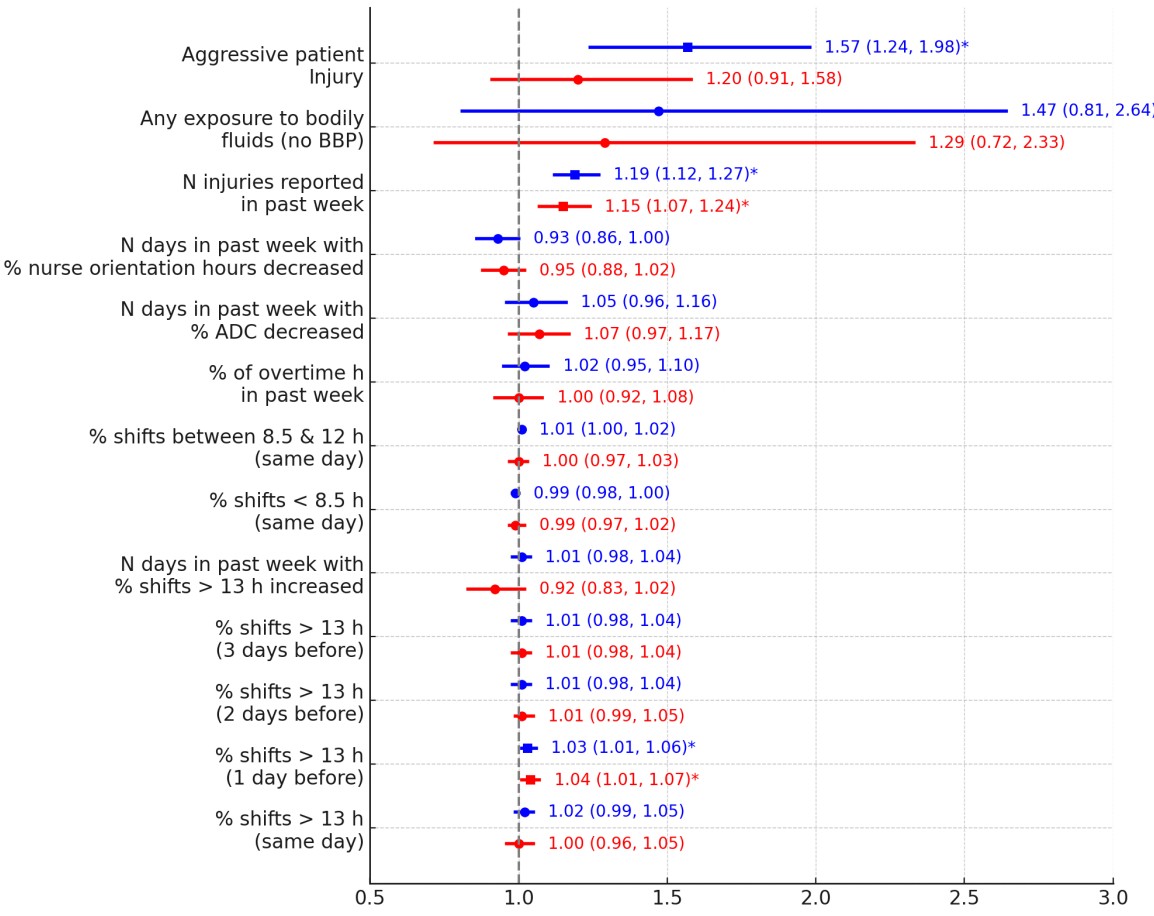

**Fig 1. Forest plot showing the odds ratios (ORs) and 95% confidence intervals (CIs) for the association between various risk factors and the likelihood of employee injury in a hospital setting (N = 7929).** The blue markers represent results from univariate models, and the red markers represent results from multivariable models adjusted for other risk factors. The vertical dashed line at OR = 1 represents no effect, squares represent significant ORs while circles represent statistically nonsignificant ORs. Significant results at 0.05 are indicated by asterisks (*). BBP: Blood-borne pathogen; ADC: Average Daily Census, N: Number.

When evaluating all predictors in the same model, only the association with the number of injuries reported in the past week remained a statistically significant predictor for injury during a particular work day, after adjusting for other risk factors (aOR:1.30, 95%CI:1.05–1.62; p = 0.0176) (Fig 2 and Table S5 in S1 File in the online supplement).

## 4. Discussion

This study evaluated unit-level and surveillance-based risk factors associated with employee injuries at a large pediatric hospital. Our findings highlight the roles of shift duration, prior injuries, patient aggression, and near-miss reporting in predicting injury risk.

Based on the analyses with only unit- and passive surveillance-based risk factors, shifts lasting longer than 13 hours on the previous day and increased percentage of shifts between 8.5 and 12 hours on a unit on the same day were associated with increased odds of employee injury, suggesting fatigue and decreased vigilance as potential underlying determinants [17]. These findings are consistent with prior research linking extended work hours and overtime with increased occupational injury risk [5,6,17–19]. While much of the existing literature focuses on adult healthcare settings, our study

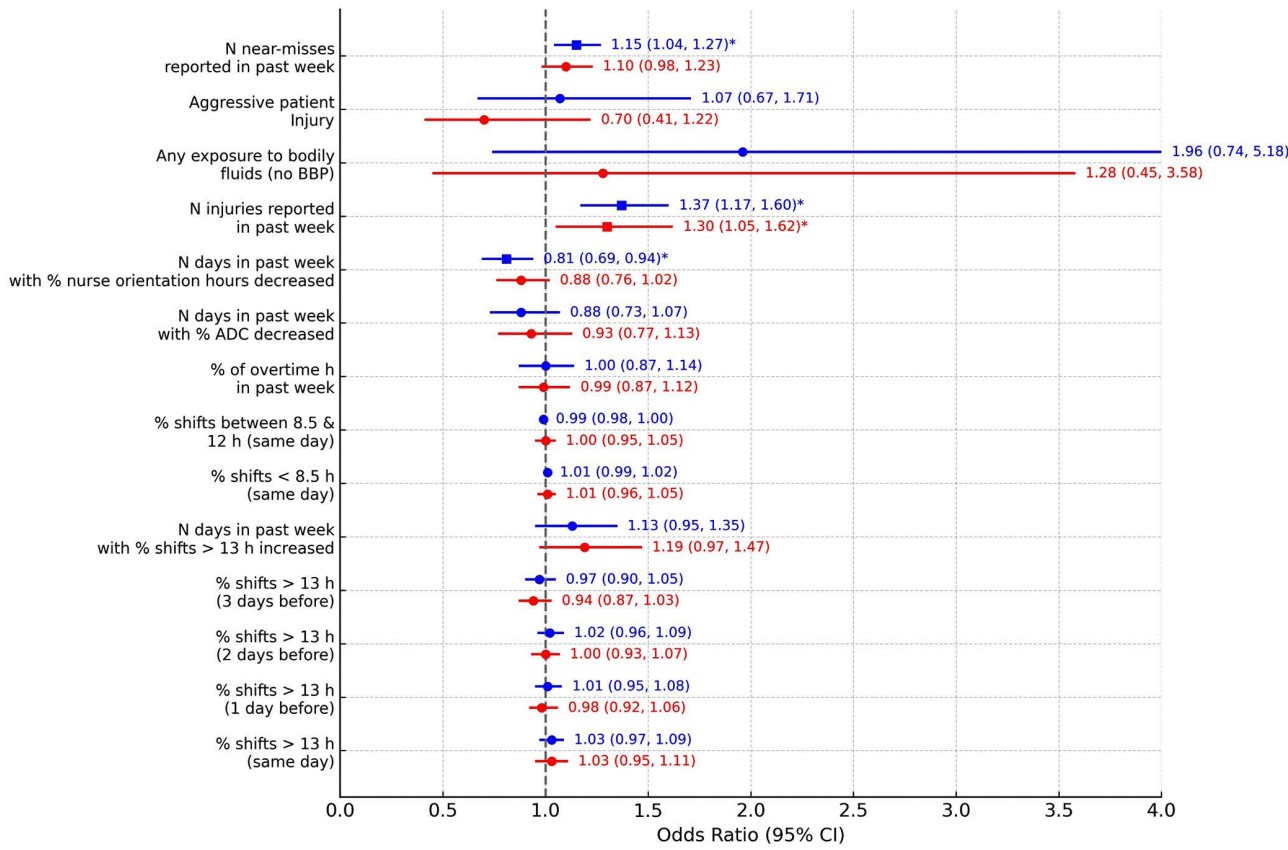

**Fig 2. Forest plot showing the odds ratios (ORs) and 95% confidence intervals (CIs) for the association between various risk factors and the likelihood of employee injury in a hospital setting.** Results from mixed effects logistic regression including active surveillance predictors (N = 1517). The blue markers represent results from univariate models, and the red markers represent results from multivariable models adjusted for other risk factors. The vertical dashed line at OR = 1 represents no effect, squares represent significant ORs while circles represent statistically insignificant ORs. Significant results at 0.05 are indicated by asterisks (*); BBP: Blood-borne pathogen; ADC: Average Daily Census, N: Number.

confirms these associations within a pediatric hospital environment. Unique aspects of pediatric care—including emotionally intensive interactions, communication challenges, and high staff-to-patient ratios—may amplify the physical and emotional demands placed on staff, highlighting the need for tailored prevention strategies in such settings.

We also found that shorter shifts (<8.5 hours) seemed to decrease injury risk. This aligns with previous research suggesting that shorter shifts can mitigate fatigue and reduce the likelihood of accidents [19]. These findings align with recent research highlighting the impact of extended shifts and insufficient recovery time on injury risk among healthcare workers. A recent study demonstrated that long work hours and inadequate rest periods are associated with an increased risk of occupational injuries, particularly musculoskeletal disorders, among hospital staff [20]. Similarly, a review found a strong association between long work hours, overtime, and an elevated risk of needlestick and sharps injuries among nurses [7]. Another analysis of healthcare worker injuries in a tertiary hospital revealed that severity of injury was strongly related to cumulative hours worked per week [21]. These studies underscore the importance of considering both shift length and cumulative exposure in efforts to prevent injuries in healthcare settings. Our results indicate that longer shifts (>13 hours) are associated with higher injury risk, while shorter shifts (<8.5 hours) appear protective. Interpreting these associations requires recognizing that different shift-length systems (e.g., 8-hour vs. > 13-hour schedules) entail different staffing structures and full-time-equivalent (FTE) requirements. Extended-shift schedules may require more FTEs to cover the same

weekly staffing needs; however, the workers assigned to >13-hour shifts experience substantially greater continuous work demands and reduced recovery time. Because our analytic unit was the unit-day rather than the individual employee, we could not directly model the number of staff physically present under each schedule. As such, our findings reflect unit-level associations between shift-length distribution and injury risk rather than per-worker risk. Even in scenarios where extended-shift systems require more personnel overall, the sustained fatigue experienced during >13-hour shifts likely remains a substantive driver of injury risk. Moreover, cumulative fatigue and lack of recovery across multiple long shifts per week likely compounds risk, consistent with prior work showing increased hazards with extended daily and weekly hours [5]. Together, these findings highlight the need for scheduling strategies that address not only total weekly work hours but also the distribution of shift lengths to minimize fatigue and injury risk.

We additionally examined the number of days in the past week when the percentage of shifts over 13 hour increased compared to the previous day, as a proxy for escalating extended shift exposure. This variable, however, was not significantly associated with injury risk. This suggests that institutional fatigue mitigation strategies or staff adaptability may buffer the effects of temporary workload surges.

The number of injuries reported in the past week was associated with increased odds of injury even after adjusting for the other unit- and passive surveillance-based risk factors. This finding is consistent with the concept of a "safety climate" according to which past incidents influence future risks, highlighting the importance of promptly addressing all reported injuries [22–25]. Also, several systematic reviews have shown that a previous injury increases the risk of future injuries [26,27]. It is important to acknowledge that the "injuries reported in the past week" variable may encompass both new incidents and ongoing consequences of prior injuries. This distinction is crucial, as ongoing injuries could contribute to the observed association with increased injury risk. Recognizing this, the variable serves as an indicator of the unit's recent safety climate and potential vulnerabilities. Future studies with more granular data distinguishing new injuries from ongoing cases would enhance the understanding of this relationship and inform targeted interventions.

While injuries from patient aggression in the past week were associated with increased odds of subsequent injuries in unadjusted analyses, this association was attenuated and became nonsignificant after adjustment for other risk factors. Additional analyses suggest that this attenuation reflects overlap with the number of injuries reported in the past week. When modeled independently, both patient aggression and prior injuries were significant predictors of injury risk; however, in the combined model, the effect of aggression was no longer statistically significant. This pattern is consistent with collinearity, but also raises the possibility that prior injuries increase the likelihood of aggressive patient behavior, placing aggression along the causal pathway between a high-risk environment and employee injury. Both variables were significant predictors of injury risk when modeled independently, but in the combined model, the effect of patient aggression was no longer statistically significant. These findings show the complexity of modeling interrelated risk factors in healthcare settings, where multiple variables may overlap conceptually. Future mediation analyses may help disentangle direct and indirect effects.

Based on the analyses with unit-based, passive surveillance, and active surveillance-based risk factors, the number of injuries reported in the past week increased the odds of injury even after adjusting for the other risk factors. The number of days in the past week when % nurses in orientation decreased from the previous day was associated with reduced odds of injury in single predictor (unadjusted) analyses but was statistically nonsignificant after adjusting for the other factors. It is noteworthy that the number of near-misses reported in the past week, an active surveillance-based risk factor, stood out as a significant predictor of injury in single-predictor analyses. Previous research has emphasized the importance of near-miss reporting as a tool for preventing future injuries [28–31]. Near-misses, by definition, are events that could have led to harm, but did not, and their recognition can serve as an early warning method. It suggests that addressing and rectifying conditions or behaviors that lead to near-misses can be a proactive strategy in injury prevention. Pham et al. demonstrated that analyzing near-miss events can provide critical insights into systemic vulnerabilities, potentially preventing actual harm [32].

In our analyses, certain factors lost statistical significance after adjusting for other risk factors, suggesting that confounding may have influenced single-predictor analyses. Additionally, the possibility of over-adjustment for collinear variables should be considered, as adjusting for highly correlated predictors can mask the true associations between individual risk factors and injury outcomes. This reflects the complexity of injury risk factors in healthcare, where multiple interrelated variables can affect the outcome of interest. The direct causes of employee injuries can be obscured by confounding factors, necessitating comprehensive analytical approaches. While multiple regression analyses can adjust for many confounders, there is always the possibility of residual confounding by risk factors not included or incorrectly measured in the study.

A limitation of our study is that it is institution-specific and while the findings are highly relevant for this hospital, they might not be generalizable to other healthcare settings with different organizational structure, patient populations, and staffing practices. To enhance the applicability of our findings, future research should involve multicenter studies encompassing diverse healthcare environments. We also relied on self-reported injury data, which may introduce reporting bias. In addition, as explained in the methods section, we did not have the unit- and day-specific denominators to properly estimate unit-and day-specific risks or odds of injury. We note that the daily denominator at risk may influence the estimation of the daily odds of risk on a unit, but is much less likely to influence the estimation of the OR within a unit, unless the risk factor is directly associated with unit size. We examined the influence of the denominator at risk by adjusting our models to the day- and unit-specific numbers of staff working on the units, and we observed negligible impact on the OR estimates. Thus, we are confident that the lack of exact denominators did not bias the association estimates. Although OSHA-recordable injuries—those resulting in death, lost workdays, restricted duty, or medical treatment beyond first aid—were captured, their rarity limited our ability to analyze them separately. Thus, our analyses focused on all reported injuries to better reflect the overall safety climate of hospital units. This approach emphasizes that both minor and severe events contribute to understanding unit-level risk and informs preventive strategies for administrators. Furthermore, our study lacked of analysis on the timing of injuries within the day or across specific shift transitions. Previous studies have identified increased injury risk associated with shift rotation, circadian rhythm disruption, and nonstandard scheduling patterns such as "flex time" (e.g., working evening followed by morning shifts) [33–35]. While these factors are intuitively linked to fatigue and physiological stress, our data were limited to the unit-day level and did not include precise timestamps for injuries or detailed shift-level scheduling. Consequently, we were unable to assess injury risk by time-of-day or shift changeover. Future studies with access to timestamped incident data and individual-level work schedules could investigate these patterns more directly and contribute to scheduling recommendations that reduce circadian strain and improve safety.

Moreover, while our study identified key risk factors for injury, the relatively small sample size for some analyses, particularly those involving active surveillance data, suggests the need for larger-scale studies. Expanding the scope of data collection to include more hospitals and employing more sophisticated analytical techniques, such as machine learning, could further refine our understanding of injury risk factors and improve predictive modeling efforts.

Although our study was based on data collected before the COVID-19 pandemic, the pandemic has undoubtedly transformed the healthcare work environment in ways that intersect with the risk factors identified here. The pandemic exacerbated staffing shortages, increased reliance on extended shifts, and introduced new physical and psychological stressors such as prolonged use of personal protective equipment (PPE), risk of infection, and emotional burnout [36,37]. These changes may have intensified pre-existing risks related to fatigue and workload, making injury prevention even more critical in the current climate. In addition, there is evidence that workplace violence and patient aggression increased during the pandemic, particularly in high-stress units like emergency and psychiatric departments [38,39]. This reinforces the importance of interventions focused on de-escalation training and support for frontline staff. While the specific context of care delivery has evolved, the core mechanisms of injury—fatigue from extended shifts, environmental stress, and inadequate staffing—remain consistent. Thus, our findings remain highly relevant and can inform safety strategies in

the post-pandemic healthcare system. Future research should examine how these risk factors have evolved in the post-COVID context and assess whether new interventions are needed to adapt to the "new normal" in healthcare operations. Longitudinal analyses comparing pre- and post-pandemic data would also help quantify the lasting effects of the pandemic on occupational injury risk.

While the post-COVID context has undoubtedly altered many aspects of healthcare delivery, it is also important to acknowledge the age of the data used in this study (2014–2017) as a limitation. Although this limits the immediacy of our conclusions, the structural risk factors we evaluated—such as long shifts, staffing strain, and patient aggression—are enduring challenges that remain highly relevant. Moreover, these data offer a unique pre-pandemic baseline against which future studies can compare patterns of injury and workplace risk. Historical data like ours provide foundational insights that help distinguish persistent from emergent risk factors over time. Future work using more recent datasets is essential to validate these patterns and to understand how systemic changes in healthcare operations—particularly those accelerated by the pandemic—have influenced injury dynamics.

The findings of this study have several important implications for healthcare management. First, hospital administrators should prioritize the development of policies that address the risks associated with extended work hours. This may include implementing maximum shift lengths, scheduling frequent breaks, and ensuring adequate staffing levels to prevent fatigue. Additionally, interventions aimed at reducing patient aggression and promoting a culture of near-miss reporting are critical for improving workplace safety.

Building upon these implications, our findings suggest several actionable and evidence-based strategies that hospitals can adopt to mitigate employee injury risk: (1) Shift Scheduling Policies: the observed association between shifts longer than 13 hours and injury risk supports limiting shift lengths to 12 hours or fewer, in line with recommendations from others [17,18,40,41]. Hospitals should also minimize mandatory overtime and incorporate adequate rest periods between shifts. Implementing predictive scheduling tools that optimize staffing based on census trends may further reduce overwork and associated risk. (2) Managing Patient Aggression: Our findings also highlight the need for proactive management of patient aggression. Recommended strategies include regular de-escalation and violence prevention training [19,42,43], use of structured risk assessments during patient intake to flag potentially aggressive behavior, and creation of rapid-response teams equipped to manage violent events. Encouraging consistent reporting of aggression-related incidents—including near-misses—followed by institutional feedback loops, can help identify unit-specific trends and guide preventive measures [32]. These approaches align with OSHA's guidelines for preventing workplace violence in healthcare and have been shown to reduce injury risk while increasing staff preparedness [17,24,32].

By proactively addressing these identified risk factors, healthcare institutions can enhance employee safety, which in turn can improve patient care outcomes. The integration of these findings into hospital safety protocols can lead to a safer work environment, reduced injury rates, and ultimately, a more effective healthcare delivery system.

## 5. Conclusions

Enhancing employee safety in healthcare settings is imperative not only for the well-being of staff but also for the quality of care provided to patients. This study has elucidated several critical risk factors that contribute to employee injuries in pediatric hospital settings. While previous studies have primarily focused on adult patient care, this study is the first to demonstrate these associations among workers in a pediatric hospital setting. By focusing on shift lengths, patient aggression, and near-miss reporting, healthcare institutions can develop targeted interventions that address these risks. Future research should continue to explore these areas, with the aim of creating safer work environments across the healthcare sector. Given the study's single-center design, future multicenter research is warranted to validate these findings across various healthcare settings and to develop comprehensive strategies for enhancing employee safety.

## Supporting information

**S1 File. Online supplementary.** This single Supporting Information file contains all supplementary materials associated with the manuscript, including: Table S1 (characteristics of employees and injury events under passive surveillance), Table S2 (sociodemographic and occupational profile of the active-surveillance cohort), Table S3 (mixed-effects logistic regression results without active-surveillance predictors), Table S4 (multivariable results excluding injuries reported in the past week), and Table S5 (mixed-effects logistic regression including active-surveillance predictors).
(DOCX)

## Acknowledgments

The authors thank all staff who contributed to data collection and analysis.

## Author contributions

**Conceptualization:** Emrah Gecili, Nancy Daraiseh, Cole Brokamp, Maurizio Macaluso.

**Formal analysis:** Emrah Gecili, Cole Brokamp, Maurizio Macaluso.

**Investigation:** Nancy Daraiseh.

**Methodology:** Emrah Gecili, Cole Brokamp, Maurizio Macaluso.

**Project administration:** Nancy Daraiseh.

**Software:** Emrah Gecili.

**Supervision:** Nancy Daraiseh, Maurizio Macaluso.

**Validation:** Emrah Gecili.

**Visualization:** Emrah Gecili.

**Writing – original draft:** Emrah Gecili, Nancy Daraiseh, Maurizio Macaluso.

**Writing – review & editing:** Emrah Gecili, Nancy Daraiseh, Cole Brokamp, Maurizio Macaluso.

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
