## [Decision Letter · Decision Letter 0]

10 Mar 2025

Dear Dr. Gecili,

Thank you for submitting your manuscript to PLOS ONE. After careful consideration, we feel that it has merit but does not fully meet PLOS ONE’s publication criteria as it currently stands. Therefore, we invite you to submit a revised version of the manuscript that addresses the points raised during the review process.

We look forward to receiving your revised manuscript.

Kind regards,

Mohamed Gamal Elsehrawy

Academic Editor

PLOS ONE

Journal Requirements:

This work was partially supported by an intramural grant by the Cincinnati Children’s Hospital Medical Center Place Outcomes Research Award. The research project that provided data from voice recordings (i.e. active injury surveillance) was funded by the Centers of Disease Control and Prevention and National Institute for Occupational Safety and Health (1R21OH010035-01A1).

This work was funded by the Centers of Disease Control and Prevention and National Institute for Occupational Safety and Health (ND, 1R21OH010035-01A1). The sponsors or funders did not play any role in the study design, data collection and analysis, decision to publish, or preparation of the manuscript. (https://www.cdc.gov/niosh/)

5. In the online submission form, you indicated that the data that support the findings of this study are confidential and not publicly available. Requests for this data can be directed to the corresponding author.

Reviewers' comments:

Reviewer's Responses to Questions

**Comments to the Author**

1. Is the manuscript technically sound, and do the data support the conclusions?

Reviewer #1: Partly

Reviewer #2: Partly

Reviewer #3: Yes

Reviewer #4: Yes

Reviewer #5: Partly

Reviewer #6: Yes

Reviewer #7: Yes

2. Has the statistical analysis been performed appropriately and rigorously?

Reviewer #1: No

Reviewer #2: Yes

Reviewer #3: Yes

Reviewer #4: I Don't Know

Reviewer #5: No

Reviewer #6: I Don't Know

Reviewer #7: Yes

3. Have the authors made all data underlying the findings in their manuscript fully available?

Reviewer #1: No

Reviewer #2: Yes

Reviewer #3: Yes

Reviewer #4: No

Reviewer #5: Yes

Reviewer #6: No

Reviewer #7: Yes

4. Is the manuscript presented in an intelligible fashion and written in standard English?

Reviewer #1: Yes

Reviewer #2: Yes

Reviewer #3: Yes

Reviewer #4: Yes

Reviewer #5: Yes

Reviewer #6: No

Reviewer #7: Yes

Reviewer #1: Weaknesses and Recommendations:

1. Ambiguities in the Methods Section:

*Data Collection Period and Scope:

- The study mentions two different data collection periods:

-- The period for collecting employee safety reports (2014–2017)

-- The participant sampling and data collection period (2013–2015)

The rationale for this discrepancy is not explained. It remains unclear whether these different data collection processes are directly comparable.

**Recommendation: The integration of these datasets should be clarified to ensure consistency and comparability.

2. Insufficient Definition of Key Variables:

*Lack of a Clear Definition for Patient Aggression:

- Although patient aggression is a key variable in the statistical analysis, its definition (e.g., physical assault, verbal threats, etc.) is not clearly stated.

**Recommendation: A precise definition of patient aggression should be provided to ensure clarity and reproducibility of the findings.

3. Potential Over-adjustment in Statistical Models:

*Collinearity Between Independent Variables:

- The significance of patient aggression disappears when previous injuries are included in the same model, which may indicate an over-adjustment issue, potentially masking the true effect.

**Recommendation: Alternative modeling approaches should be considered to examine the independent effects of these variables. For instance, separate models could be tested for patient aggression and previous injuries to assess their individual contributions.

4. Sample Size and Generalizability Concerns:

*Single-hospital Sample:

- The study is based on data from a single pediatric hospital, which may limit the generalizability of the findings to other healthcare settings.

**Recommendation: The study’s limitations should be explicitly stated, and the need for future multi-center studies should be emphasized to improve external validity.

5. Limitations in Data Sharing:

*Restricted Data Access:

- The statement "The data that support the findings of this study are confidential and not publicly available." may raise concerns about the transparency and reproducibility of the study.

**Recommendation: If possible, summary statistics or anonymized datasets should be made available to enhance transparency and facilitate replication of the findings.

Reviewer #2: Unfortunately, I cannot recommend publication of the article due to several major shortcomings in both its design and its relevance and methodological rigor.

Temporal relevance of the data: Data collection in the study was conducted 10 years ago. This temporality presents a significant limitation since the results and conclusions of the study may have lost relevance in the current nursing context. Health and nursing research is constantly evolving, and old data may not reflect current conditions, needs, and practices in the discipline. This weakens the external validity of the study and its applicability to contemporary practice.

Few references and lack of updating: The article presents only 18 bibliographic references, and the most recent ones date from 2020. This limited number of references and the lack of updating in the bibliographic sources evidence a deficiency in the integration of the most recent advances in the scientific literature on the subject. This calls into question the completeness of the theoretical framework and the state of the art of the study, which is crucial to ensure that the research is well grounded in the most recent developments in the discipline.

Insufficiently detailed methodology: Although the study mentions that it uses mixed methods, no clear explanation of how the qualitative part of the research was conducted is provided. Nor is the qualitative part stated in the results.

Lack of information on the number of participants: The methodology does not specify the number of participants involved in the study. This omission is a serious problem since the number of participants is an essential factor in assessing the validity and reliability of the results. Without this information, it is impossible to judge the representativeness of the sample or the statistical power of the analyses, which directly affects the robustness of the findings.

Absence of reference to EQUATOR guidelines: The article does not explicitly mention which good practice guidelines and recommendations, such as those of EQUATOR, were followed during the design and execution of the study. The EQUATOR guidelines are fundamental to ensure that research is conducted following international ethical and methodological standards. The omission of this information suggests that the study may not have adhered to best practices in terms of transparency and methodological rigor.

Given the combination of these factors, I believe that the manuscript does not meet the minimum requirements of quality and relevance necessary to be published in a scientific nursing journal. I suggest that major revisions be made in terms of methodology, updating of references, and temporal relevance of the data before it is considered for publication.

Reviewer #3: This study investigates risk factors for employee injuries in a pediatric hospital setting through analysis of self-reported injury data, active surveillance via voice recording, and hospital unit measures. Using mixed effects logistic regression models, the authors identified several significant risk factors, including shifts exceeding 13 hours, patient aggression, and prior injury occurrences.

This research addresses a critical real-world issue with significant implications. Identifying risk factors for hospital worker injuries is essential for improving healthcare workplace environments and employee wellbeing.

However, there are several points author need to consider:

1. The authors have not provided essential descriptive information about the sample population. Details regarding the participants' gender, age, education level, job roles, and work experience would provide important context for interpreting the results and understanding the generalizability of the findings.

2.While the authors mention that their findings can "guide the development of targeted interventions," the manuscript would benefit from a more detailed discussion of specific strategies hospitals could implement based on these findings. For example, how might hospitals modify shift scheduling policies or address patient aggression more effectively?

3.The study appears to rely on pre-pandemic data, which limits its contemporary relevance. Given the significant impact of COVID-19 on healthcare work environments, the authors should include a discussion of how the pandemic may have altered risk factors and working conditions for healthcare employees, and how their findings might translate to the post-pandemic context.

4. The authors should provide more comprehensive information about their data processing methods, particularly regarding how they coded and analyzed the voice recordings from active surveillance. This additional methodological transparency would strengthen the validity and reproducibility of the research.

Reviewer #4: Thank you for the opportunity to review this work.

With this, I have a few questions:

For methods, I am curious as to the nature of the dataset in general (given that it isn't included in the supplemental materials). Typically for a study of this nature, the dataset itself is usually described (e.g. 'x' number of survey responses, with 'y' being excluded for 'z' reason, resulting in 'c' subjects left for analysis...). I would also indicate that this is a retrospective review of prospectively collected data (which is something I am assuming per line 167 in the manuscript). As such, there are usually variables that are missing in such data, and I would be interested to know if that resulted in subjects / responses being excluded.

In any case, I presume that the total population of survey results you are analyzing is 9446 (7929 without active surveillance predictors and 1517 with, per Table 1). My confusion stems from whether or not the with/without active surveillance predictor datasets are supposed to represent a case/control type analysis. And I presume the intent was to compare data from when the employees were encouraged to include near-misses in reporting, though the manuscript doesn't make clear what the exact difference between "with active surveillance" and "without active surveillance" means in terms of the study population.

Along with this, there does not appear to be any evaluation as to time-of-day an injury might occur. Correlating injury to time-of-shift may be helpful, as other papers have noted that "flex time" shifts (evening followed by daytime, for example) are a potential risk factor for injury. Given changes in circadian rhythm, this makes intuitive sense, but is not addressed in this paper.

In addition, although you show a forest plot with an indication that there is a significant (but small) odds ratio increase for longer shifts, there does not seem to be a dose-response to such ratios - which seems counter-intuitive if fatigue is the mechanism by which these injuries are thought to occur. Most notably, the percentage increase of over 13-hour shifts the same day did NOT show a corresponding statistically significant increase in reported injuries, similar to the lack of change with the percent (number?) shifts over 13h in past week. (I am not clear as to if the "N days in past week with % shifts > 13 h increased" refers to the percentage of shifts over 13 hours in a week, or the number of over-13 hour days in a given week as the variable).

As it stands, the data you are presenting suggest that scheduling 13-hour shifts increase injuries, while scheduling <8.5 hour shifts can reduce injuries - though that notion intuitively doesn't help when trying to schedule a provider for, as an example, three 13-hour shifts per week vs 5 or 6 8.5-hour shifts per week, as the at-risk work time exposure per individual worker is different.

On that note, I do wonder how your regression analysis was set up. Reviewing the tables, it's evident that the number of shifts is entered as a percentage variable, while the number of "days in past week" for three variables are set as an integer ("n"), along with injuries reported and near misses reported. Was there a reason to use "percentage" over integer values when noting numbers of shift lengths? If integer values were used, the conclusion would theoretically read "for every individual shift worked by unit staff over 13h there is an increased 'x' risk for injury", which is easier to understand. Conversely, a tighter association with fatigue might be present if "hours worked by staff 7 days prior" substituted for the shift variables - or at least "number of >13h shifts worked by unit staff in 7 days prior". I would argue that such dependent variables would be better, as they are not

I also note that your manuscript lists "percent NHPPD" and "Unit RN Vacancy" as potential risk factors. Those variables are not visibly noted in your regression. Is there a reason why these were left off?

I would also be interested to know how the data analysis was performed (e.g. SASS, STATA?). Your data collection and database analysis method isn't listed in the manuscript. Also, the use of a mixed-effect logistic regression was described, however the statistical analysis used in the univariate calculation of "injury from patient aggression" (line 251-252) was not described.

Finally, the issue of relevance to healthcare administrators with regards to this paper comes into question. Namely, is there data regarding the severity of the injuries encountered? One can argue that administrators would be more concerned about the long-term risk for injury, as it would be different between a potential blood-borne contact with no long-term sequelae vs an assault that resulted in worker-days lost. Given the nature of this analysis, my supposition is that the number of injuries sustained that were severe enough to result in a lost day at work were not enough to analyze properly - though I do wonder if the reason "injury reported in past week" variable was significant because it reflected a continuation of an injury sustained by a staff member the week prior.

As an aside, most trauma and injury prevention literature already utilize "ISS" as an abbreviation for "injury severity score". As this paper does not list the severity of injuries, rather than document the self-reported existence or absence of such, I'd suggest that your abbreviation should be changed to "IISS", for "institutional injury surveillance system", in order to avoid confusion.

As for formatting, including a legend embedded within your figures (vs having them within the body of the manuscript - see lines 234-240, for example) would be helpful. And is there a reason that your table of model results are only within the supplemental material, rather than as a table within the main manuscript?

Reviewer #5: A great article !

I have some reservations

Regarding the discussion, it needs to be more elaborate and please mention study populations. As this study dealt with pediatric.

Several factors differ.

As mentioned you have active and passive surveillance factors, I would want a bit of explanation on this regard.

on 224 line “unadjusted analyses”), a 1% = Bracket just appeared closed end, while there was no open ended.

Table 1 needs more explanation

Reviewer #6: This manuscript reports a study that aims to explore and quantify the risk factors associated with employee injuries on inpatient units. It appears that an additional aim was to explore the value of adding an active approach to incident surveillance – that of carrying a pocket voice recorder to more proactively capture near misses.

The study analyzed employee injury data self-reported to the institutional surveillance system, as well as incident reports collected from a sample of employees through active surveillance (voice recording), and hospital unit-based measures of patient density and employee workload as potential determinants of injury

The main findings were that shifts of >13 hours on the prior day and patient aggression are positively associated with injury likelihood and shifts <8.5 hours on the same day are negatively associated.

Overall, the work reported covers an important topic, reports new information from the paediatric setting, and appears to be a well conducted study.

Some sections of writing are too lengthy, omit important information, or are in the wrong section. Referencing is too thin throughout, particularly in the Introduction. Attention is also needed in the Discussion. Suggestions for improving the clarity and organisation of writing are given below.

Table 1 is not easy to follow and needs review of how information is presented to make clear what it is, and clean up a few inconsistencies. Statistical review is recommended.

Introduction

References are too thin throughout. All statements of fact need to be substantiated with evidence.

An introduction of the different types of incident surveillance system data collection methods (e.g., online, call centre, pocket recorder) etc is needed given the exploration of this aspect in the study.

Lines 102-119 are excessively wordy. Any justification for the study needs to be presented earlier in the section. Any postulation about the meaning and impact of the study needs to go in the Discussion. These paragraphs should be reduced to max 1-2 short sentences that state the study aim(s). Details of the methods should go in the Methods section.

Methods

Start by stating the study design. Identify a study reporting guideline appropriate to the study design, review your manuscript against this and attach the check list as an appendix, and note which guideline you used in the first paragraph of the methods.

Make clear that the study is conducted in a single hospital. (Mentioned in the aims at end of previous section – suggest move to Methods).

Clearer introduction of inclusion/exclusion criteria is needed. Definitions of some of the variables is needed. For example, how is ‘patient aggression’ defined and measured?

Regarding data collection, clearer reporting is needed of how data were extracted, from what systems in what format, by whom.

Line 167-76: There appears to be a staff population subgroup with separate method of data collection for a short period within the study timeframe. Is this the ‘Active surveillance’ data? Suggest give this a sub-heading. Introduce and justify. Please briefly describe the larger study from which this group was drawn and how potential participants for this study were identified and by whom they were invited. Clarify how this data is analysed.

Results

It would be helpful if the total number of staff, hours worked, and number of units from which data was drawn could be reported in the opening of the Results section.

Line 210-4: Judgement, opinion and other interpretation should be reserved for the discussion. Recommend remove from here ‘Notably’ and full sentence of Lines 212-4.

Table 1: Some of the values are presented in variable format and it is not clear what the % represent – i.e., of what total. Please review for clarity and completeness.

Discussion

Lines 305-8: A suggestion is for this sentence to form the beginning of the next paragraph

Line 323: Needs referencing

Reviewer #7: I commend the authors for their thorough work investigating the risk factors associated with hospital employee injuries. The authors effectively contextualized the present issue and described the need for this research. Clear descriptions of hospital injury procedures, variables of interest, and data analysis methodology are presented. The authors captured near-misses along with recorded injuries, which provides information not typically investigated when studying workplace safety. Potential over-adjustment of the statistical model and collinearity between variables were adequately addressed. Tables and figures provide are clear and provided all relevant data. Overall results demonstrate the importance of employee shift duration and prompt responses to injuries for employee safety. Based on their findings, the authors suggested several areas requiring improvements to healthcare management and administration to enhance employee safety.

A few minor suggestions are provided below to enhance clarity.

1. For clarity regarding the five metrics assessing risk factors for employees, the authors are recommended to provide more precise relationships between the measure and what the measure indicates. This was done for metrics 1, 2, and 3, however, readers may benefit from clearer descriptions for metrics 4 and 5. For metric 4, the authors are suggested to specify the value that may indicate decreased skill level or experience. For metric 5, the authors are suggested to specify that higher values may be an indicator of staff fatigue. These changes will improve clarity for readers.

2. In the results sections, the authors are recommended to specify variable units of measurement to clarify values for the reader. Although clearly described in the methods section, there is limited mention that measures were captured at the unit level. For example, in line 263–264, the authors could include that the number of near-misses were recorded for each unit (e.g., “The average number of near-misses reported in the past week was 0.65 (SD 1.53) per unit in the active surveillance dataset”). This addition would clarify the results for readers who may skip the methods section.

3. Table 1 has some inconsistent formatting and unnecessary characters that can be removed.

a. Percentages are not included in the table consistently across variables, which may confuse readers. For example, the categorical variable, Injury Occurrence, includes a percentage sign within the parentheses to denote that 477 is 6.02% of the 7929 participants included in the study. However, % shift > 13 h provides continuous data, but the use of a percent sign within the parentheses impedes reader understanding of whether the values within the parentheses for this variable are SDs for a mean or percentages for n cases. With percentages and Ns specified in the variables column and the table footnote providing additional information, the authors are recommended to remove the percentages from the table columns containing data.

b. Similarly, the use of “days” in the data columns (e.g., for the variable N days in the past week when % shifts >13 h increased over the previous day) is not necessary and can be removed as it adds noise to the table. The “days” qualifier has already been stated in the Variables column, which provides sufficient information for each variable.

c. The authors are recommended to write out the ADC acronym in the footnote.

These changes would improve readability as well as consistency with the supplemental table.

Overall, the authors provided novel data regarding hospital employee injury risk factors. Results were adequately presented and the study findings are adequately interpreted. Author concerns and study limitations were appropriately addressed throughout the manuscript and yielded opportunities to discuss the complexity of this field of research.

**Do you want your identity to be public for this peer review?** For information about this choice, including consent withdrawal, please see our Privacy Policy

Reviewer #1: No

Reviewer #2: No

Reviewer #3: No

Reviewer #4: **Yes: ** Julius D Cheng, MD MPH

Reviewer #5: No

Reviewer #6: **Yes: ** Miranda Buhler

Reviewer #7: **Yes: ** Sorina Andrei

---

## [Author Response · Author response to Decision Letter 1]

15 Oct 2025

Point-by-point responses to reviewers

Manuscript ID: PONE-D-25-05110

***Denotes authors’ point-by-point responses

We sincerely thank the Associate Editor and Reviewers for their thoughtful assessment and feedback, which have helped improve our manuscript. Below, we provide our responses and refer to the tracked changes version, where additions and deletions are marked accordingly. Below, our responses are indicated with *** and presented in italics.

Reviewer #1:

R1.1: Ambiguities in the Methods Section:

*Data Collection Period and Scope:

- The study mentions two different data collection periods:

-- The period for collecting employee safety reports (2014–2017)

-- The participant sampling and data collection period (2013–2015)

The rationale for this discrepancy is not explained. It remains unclear whether these different data collection processes are directly comparable.

**Recommendation: The integration of these datasets should be clarified to ensure consistency and comparability.

***Authors’ response: Thank you for your comment. The decision to integrate both passive and active injury surveillance datasets was intentional and central to our study’s objectives. Passive surveillance, while efficient and broad in scope, often underreports injuries due to its reliance on existing records and reporting mechanisms. In contrast, active surveillance involves proactive data collection and tends to capture a more accurate picture of injury occurrences. Merging these datasets allows for a more comprehensive evaluation of employee injuries. This clarification has been added to section 2.1 Data Collection.

R1.2: Insufficient Definition of Key Variables:

*Lack of a Clear Definition for Patient Aggression:

- Although patient aggression is a key variable in the statistical analysis, its definition (e.g., physical assault, verbal threats, etc.) is not clearly stated.

**Recommendation: A precise definition of patient aggression should be provided to ensure clarity and reproducibility of the findings.

***Authors’ response: Thank you for your comment. Indeed, this concern is related to a similar comment by Reviewer 6. Please see comment R6.7 and our response below. Patient aggression was defined per institution policy as any aggressive/violent behaviors to others and verbal aggression/threats/agitation. This has been added to section 2.1 Data Collection.

R1.3: Potential Over-adjustment in Statistical Models:

*Collinearity Between Independent Variables:

- The significance of patient aggression disappears when previous injuries are included in the same model, which may indicate an over-adjustment issue, potentially masking the true effect.

**Recommendation: Alternative modeling approaches should be considered to examine the independent effects of these variables. For instance, separate models could be tested for patient aggression and previous injuries to assess their individual contributions.

***Authors’ response: We thank the reviewer for this important observation. We agree that the diminished significance of the patient aggression variable when controlling for prior injuries may reflect over-adjustment due to collinearity between the two predictors. To address this, we conducted additional analyses to assess their individual and joint contributions to injury risk. Assessment of Collinearity: We examined the Pearson correlation between the number of injuries in the past week and patient aggression, which was r = 0.66, indicating a strong positive relationship. To more formally assess collinearity in the regression models, we also inspected the covariance/correlation matrix of parameter estimates and calculated variance inflation factors (VIFs). VIF values for most predictors, including patient aggression (VIF = 1.32) and number of injuries (VIF = 1.39), were low, suggesting that collinearity between these two predictors was modest. Overall, these results indicate that while multicollinearity between patient aggression and injury counts is present, it is not severe; nonetheless, their conceptual and statistical overlap may contribute to attenuation of the aggression effect when both are included in the same model.

Alternative Modeling Approaches: To assess the potential for over-adjustment, we estimated three models:

• Model 1: Includes only aggressive patient injury as a predictor. In this model, aggressive patient injury was a significant predictor (OR = 1.53, 95% CI: 1.21–1.95).

• Model 2: Includes both aggressive patient injury and number of injuries in the past week. In this model, the effect of aggressive patient injury is no longer statistically significant (OR = 1.20, 95% CI: 0.91–1.58), while past week injuries remain significant (OR = 1.15, 95% CI: 1.07–1.24).

• Model 3: Includes only number of injuries in the past week. In this model, past injuries remain a robust predictor of injury risk (OR = 1.18, 95% CI: 1.10–1.26).

These results support the reviewer’s concern: when both predictors are included, the effect of aggressive patient injury is attenuated, likely due to overlap with recent injury history. Although VIF values for both predictors were low (~1.3–1.4), their strong correlation (r = 0.66) and conceptual relatedness suggest over-adjustment. Alternative model comparisons showed that aggression alone was a significant predictor, past injuries alone were also robust, but when combined, past injuries dominated and the aggression effect lost significance. Importantly, we now also acknowledge that increased frequency of injuries in a unit may itself elevate the risk of subsequent aggressive patient behavior, placing aggression on the causal pathway between a high-risk environment and employee injury. In this case, the two variables cannot be considered fully independent covariates, and future work using mediation analysis will be needed to disentangle direct and indirect effects.

Conclusion: We have revised the manuscript to note that patient aggression and past injury counts are not only correlated but may also be causally linked. We clarify that each predictor independently associates with injury risk, but joint inclusion likely masks the aggression effect because of this causal overlap. Additional explanation has been added to the Results and Discussion to highlight this possibility and point toward mediation analysis as a more appropriate framework for future work. Please see page 15 and lines 321-334 and page 19 and lines 413-423.

R1.4: Sample Size and Generalizability Concerns:

*Single-hospital Sample:

- The study is based on data from a single pediatric hospital, which may limit the generalizability of the findings to other healthcare settings.

**Recommendation: The study’s limitations should be explicitly stated, and the need for future multi-center studies should be emphasized to improve external validity.

***Authors’ response: We appreciate the reviewer's observation regarding the limitation of our study being conducted at a single pediatric hospital, which may affect the generalizability of our findings. We acknowledge that single-center studies can limit external validity due to factors such as unique institutional practices, patient demographics, and staffing models. To address this concern, we edited the Discussion and Conclusions sections. Please see page 21 and lines 449-453 and page 25 and lines 540-543.

R1.5: Limitations in Data Sharing:

*Restricted Data Access:

- The statement "The data that support the findings of this study are confidential and not publicly available." may raise concerns about the transparency and reproducibility of the study.

**Recommendation: If possible, summary statistics or anonymized datasets should be made available to enhance transparency and facilitate replication of the findings.

***Authors’ response: Thank you for your comment. We appreciate the reviewer's concern regarding the transparency and reproducibility of our study, particularly in relation to data availability. We acknowledge the importance of providing access to data that support research findings. However, due to the sensitive nature of the data involved in our study, which includes identifiable employee information and confidential institutional records, we are unable to make the full dataset publicly available. To address this concern, we have revised our data availability statement.

Reviewer #2:

Unfortunately, I cannot recommend publication of the article due to several major shortcomings in both its design and its relevance and methodological rigor.

***Authors’ response: We respectfully disagree with the reviewer's assessment. We have tried to address the specific comments and hope that the revised manuscript is acceptable for publication.

R2.1: Temporal relevance of the data: Data collection in the study was conducted 10 years ago. This temporality presents a significant limitation since the results and conclusions of the study may have lost relevance in the current nursing context. Health and nursing research is constantly evolving, and old data may not reflect current conditions, needs, and practices in the discipline. This weakens the external validity of the study and its applicability to contemporary practice.

***Authors’ response: We thank the reviewer for this important observation. We fully acknowledge that the temporal distance from data collection to publication represents a limitation that could influence the generalizability and external validity of our findings. However, we respectfully argue that the core structural and organizational risk factors examined—such as extended shifts, staffing shortages, nurse orientation levels, and exposure to patient aggression—remain persistent challenges in healthcare, including pediatric inpatient settings. The mechanisms linking these workplace conditions to injury risk continue to be highly relevant, even as specific contextual features of clinical care evolve.

Moreover, the insights from this historical data provide a baseline perspective that can inform understanding of how such risk factors operated before the transformative disruptions introduced by the COVID-19 pandemic. We have revised the Discussion section to explicitly address this temporal limitation and position the findings as both historically informative and foundational for comparison with more recent studies. Please see pages 23-24 and lines 498-507.

R2.2: Few references and lack of updating: The article presents only 18 bibliographic references, and the most recent ones date from 2020. This limited number of references and the lack of updating in the bibliographic sources evidence a deficiency in the integration of the most recent advances in the scientific literature on the subject. This calls into question the completeness of the theoretical framework and the state of the art of the study, which is crucial to ensure that the research is well grounded in the most recent developments in the discipline.

***Authors’ response: In response, we have conducted a comprehensive review of the most recent scientific literature published between 2021 and 2025 that is relevant to occupational injuries in healthcare settings, fatigue and shift work, and injury prevention.

We identified several recent studies and reviews that have reinforced the importance of factors such as extended shifts, rotating schedules, and cumulative workload in relation to workplace injury risk among healthcare workers.

We have revised the Introduction, Discussion, References sections to incorporate these recent studies.

R2.3: Insufficiently detailed methodology: Although the study mentions that it uses mixed methods, no clear explanation of how the qualitative part of the research was conducted is provided. Nor is the qualitative part stated in the results.

***Authors’ response: We thank the reviewer for this observation. We would like to clarify that although our study involved the use of voice recordings, which were qualitative in form, the objective was not to conduct a qualitative thematic analysis or a traditional mixed methods study as typically defined in health services research (i.e., combining qualitative and quantitative research questions, methods, and analyses). Instead, we used the voice-recorded reports to generate structured, coded variables (e.g., occurrence of near-misses) for inclusion in our quantitative modeling. To avoid confusion, we note that the term “mixed models” in the manuscript refers to regression models with both fixed and random effects, not to mixed methods research.

R2.4: Lack of information on the number of participants: The methodology does not specify the number of participants involved in the study. This omission is a serious problem since the number of participants is an essential factor in assessing the validity and reliability of the results. Without this information, it is impossible to judge the representativeness of the sample or the statistical power of the analyses, which directly affects the robustness of the findings.

***Authors’ response: We thank the reviewer for this important observation. We have clarified the study populations and analytic units for both data sources.

The passive surveillance dataset included 7,929 unit-days of aggregated surveillance and operational data collected across 25 inpatient units, encompassing approximately 1,214 employees from intensive care, medical–surgical, and psychiatry services.

The active surveillance dataset contributed 1,517 unit-days derived from 607 employees who participated in a time-limited observational study where staff recorded injuries and near-misses using voice recorders (see Table S1 and Table S2). Because analyses were conducted at the unit-day level, the associations detected represent relationships between unit-level exposures and injury risk rather than individual-level effects. We have clarified these sample sizes and aggregation levels in both the Methods (Data Sources) and Results (Opening paragraph) sections for transparency.

R2.5: Absence of reference to EQUATOR guidelines: The article does not explicitly mention which good practice guidelines and recommendations, such as those of EQUATOR, were followed during the design and execution of the study. The EQUATOR guidelines are fundamental to ensure that research is conducted following international ethical and methodological standards. The omission of this information suggests that the study may not have adhered to best practices in terms of transparency and methodological rigor.

Given the combination of these factors, I believe that the manuscript does not meet the minimum requirements of quality and relevance necessary to be published in a scientific nursing journal. I suggest that major revisions be made in terms of methodology, updating of references, and temporal relevance of the data before it is considered for publication.

***Authors’ response: We appreciate the reviewer’s emphasis on best practice guidelines. We would like to clarify that not explicitly stating adherence to such standards in the original manuscript does not imply that they were not followed. This study was designed and reported in accordance with international best practice guidelines, specifically STROBE (Strengthening the Reporting of Observational Studies in Epidemiology), which is the appropriate EQUATOR guideline for observational studies. We have now updated the Methods section to explicitly state this adherence and ensured that all relevant checklist items are addressed in the manuscript.

We also recognize the importance of temporal context and have added text clarifying the impact of the COVID-19 pandemic and changes in healthcare delivery since the data collection period. In addition, we have updated our reference list to reflect recent literature and methodological developments.

Reviewer #3:

This study investigates risk factors for employee injuries in a pediatric hospital setting through analysis of self-reported injury data, active surveillance via voice recording, and hospital unit measures. Using mixed effects logistic regression models, the authors identified several significant risk factors, including shifts exceeding 13 hours, patient aggression, and prior injury occurrences.

This research addresses a critical real-world issue w

---

## [Decision Letter · Decision Letter 1]

6 Nov 2025

We look forward to receiving your revised manuscript.

Kind regards,

Mohamed Gamal Elsehrawy

Academic Editor

PLOS ONE

**Journal Requirements:**

Reviewers' comments:

Reviewer's Responses to Questions

**Comments to the Author**

Reviewer #3: All comments have been addressed

Reviewer #4: All comments have been addressed

2. Is the manuscript technically sound, and do the data support the conclusions?

Reviewer #3: Yes

Reviewer #4: Yes

3. Has the statistical analysis been performed appropriately and rigorously?

Reviewer #3: Yes

Reviewer #4: I Don't Know

4. Have the authors made all data underlying the findings in their manuscript fully available?

Reviewer #3: Yes

Reviewer #4: Yes

5. Is the manuscript presented in an intelligible fashion and written in standard English?

Reviewer #3: Yes

Reviewer #4: Yes

**Reviewer #3:**  The authors have adequately addressed the concerns raised in the previous round of review. The revisions have improved the clarity and rigor of the manuscript. I have no further comments at this stage.

**Reviewer #4: ** Thank you for your responses to my concerns.

I would note that for the response re: R4.5 - "Thus, the observed associations may in fact be conservative: when more employees work shorter shifts, the denominator of workers at risk increases, yet the overall injury risk remains lower."

I'd argue that the converse is true - if the standard work-week is 40 hours, on a 10-RN unit (for example), an 8-hour shift system would require 42 FTE's to staff the 210 slots fully for one week, without overtime.

Conversely, a "39-for-40" system with 13-hour shifts (presuming half-hour overlap at each end) would require 47 nurses to cover the same unit over 140 slots, if there's no overtime. So there are more FTE's in the 12-hour-shift scenario needed vs the 8-hour.

**Do you want your identity to be public for this peer review?** For information about this choice, including consent withdrawal, please see our Privacy Policy

Reviewer #3: No

Reviewer #4: **Yes: ** Julius D. Cheng, MD MPH

---

## [Author Response · Author response to Decision Letter 2]

19 Nov 2025

We sincerely thank the Associate Editor and Reviewers for their thoughtful assessment and feedback, which have helped improve our manuscript. Below, we provide our responses and refer to the tracked changes version, where additions and deletions are marked accordingly. Below, our responses are indicated with *** and presented in italics.

Reviewer #3: The authors have adequately addressed the concerns raised in the previous round of review. The revisions have improved the clarity and rigor of the manuscript. I have no further comments at this stage.

***Authors’ response: We thank Reviewer #3 for the positive evaluation and appreciate the acknowledgement of our revisions. We are glad the updates improved the clarity and rigor of the manuscript.

Reviewer #4: Thank you for your responses to my concerns.

I would note that for the response re: R4.5 - "Thus, the observed associations may in fact be conservative: when more employees work shorter shifts, the denominator of workers at risk increases, yet the overall injury risk remains lower."

I'd argue that the converse is true - if the standard work-week is 40 hours, on a 10-RN unit (for example), an 8-hour shift system would require 42 FTE's to staff the 210 slots fully for one week, without overtime.

Conversely, a "39-for-40" system with 13-hour shifts (presuming half-hour overlap at each end) would require 47 nurses to cover the same unit over 140 slots, if there's no overtime. So there are more FTE's in the 12-hour-shift scenario needed vs the 8-hour.

***Authors’ response: We thank the reviewer for this helpful clarification. We agree that, from a scheduling perspective, different shift-length systems (e.g., 8-hour vs. >13-hour extended shifts) require different full-time-equivalent (FTE) staffing configurations. As the reviewer illustrates, a schedule built on >13-hour shifts may require more individual staff than an 8-hour system to ensure full weekly coverage. We have revised the Discussion to reflect this important point and to clarify how it relates to our analytic approach.

At the same time, our analytic unit was the unit-day—not the individual worker—so we could not directly model the number of staff physically present or the underlying FTE structure associated with each scheduling system. This means our results reflect unit-level associations between the distribution of shift lengths and injury risk, rather than per-worker risk.

Importantly, even if >13-hour systems sometimes require more staff overall, the workers assigned to these extended shifts experience substantially greater fatigue and reduced recovery time than those working shorter shifts. This aligns with extensive evidence showing that extended daily hours—particularly >12–13 hours—are associated with elevated injury risk due to cumulative fatigue, sleep debt, and decreased vigilance. We have revised the Discussion to incorporate these clarifications. Please see pages 18-19 and lines 389-399.

---

## [Decision Letter · Decision Letter 2]

2 Dec 2025

Hospital Unit Working Conditions and Risk for Employee Injury

PONE-D-25-05110R2

Dear Author,

We’re pleased to inform you that your manuscript has been judged scientifically suitable for publication and will be formally accepted for publication once it meets all outstanding technical requirements.

Kind regards,

Mohamed Gamal Elsehrawy

Academic Editor

PLOS ONE

Additional Editor Comments (optional):

Reviewers' comments:

Reviewer's Responses to Questions

**Comments to the Author**

Reviewer #3: All comments have been addressed

Reviewer #4: All comments have been addressed

2. Is the manuscript technically sound, and do the data support the conclusions?

Reviewer #3: Yes

Reviewer #4: Yes

3. Has the statistical analysis been performed appropriately and rigorously?

Reviewer #3: Yes

Reviewer #4: I Don't Know

4. Have the authors made all data underlying the findings in their manuscript fully available?

Reviewer #3: No

Reviewer #4: Yes

5. Is the manuscript presented in an intelligible fashion and written in standard English?

Reviewer #3: Yes

Reviewer #4: Yes

Reviewer #3: The authors have properly addressed my concerns and comments and I don't have any further comments for this piece.

Reviewer #4: I appreciate the response with regards to staffing, thank you. Perhaps additional information with regards to specific "dose limits" as to work hours will show the actual optimal shift work-hour length in comparison to number of hand-off errors from a patient-safety standpoint.

**Do you want your identity to be public for this peer review?** For information about this choice, including consent withdrawal, please see our Privacy Policy

Reviewer #3: No

Reviewer #4: No

---

## [Editor Report · Acceptance letter]

PONE-D-25-05110R2

PLOS One

Dear Dr. Gecili,

I'm pleased to inform you that your manuscript has been deemed suitable for publication in PLOS One. Congratulations! Your manuscript is now being handed over to our production team.

Kind regards,

on behalf of

Dr. Mohamed Gamal Elsehrawy

Academic Editor

PLOS One